# Comparison of Combustion and Pyrolysis Behavior of the Peanut Shells in Air and N$_2$: Kinetics, Thermodynamics and Gas Emissions

**Zhenghui Xu [1,2,3], Xiang Xiao [1,3], Ping Fang [1,3,*], Lyumeng Ye [4], Jianhang Huang [1,3], Haiwen Wu [1,3], Zijun Tang [1,3] and Dongyao Chen [1,3]**

[1]  South China Institute of Environmental Sciences, Ministry of Ecology and Environment, Guangzhou 510655, China; xuzhenghui_ncu@163.com (Z.X.); xiaoxiang@scies.org (X.X.); huangjianhang@scies.org (J.H.); 13570049350@163.com (H.W.); tangzijun@scies.org (Z.T.); chendongyao@scies.org (D.C.)

[2]  Key Laboratory of Poyang Lake Environment and Resource Utilization, Ministry of Education, School of Resources Environmental and Chemical Engineering, Nanchang University, Nanchang 330031, China

[3]  The Key Laboratory of Water and Air Pollution Control of Guangdong Province, Guangzhou 510655, China

[4]  School of Environmental Science and Engineering, Sun Yat-sen University, Guangzhou 510655, China; yelm7@mail2.sysu.edu.cn

*  Correspondence: fangping@scies.org

**Abstract:** The influences of four heating rates on the combustion and pyrolysis behavior in the N$_2$ and air atmosphere were investigated by the Fourier transform infrared spectrometry (FTIR) and thermogravimetric (TG) analysis.  the distributed activation energy model (DEAM) and Flynn-Wall-Ozawa (FWO) were used to estimate *Ea* and *A*, Δ*H*, Δ*G* and Δ*S*. Experimental results showed that the similar thermal behavior emerged, but the temperatures in the air and N$_2$ atmospheres representing the end of the reaction were about 500 °C and 550 °C, respectively. The results of FTIR showed the peak positions were basically the same, but the concentrations of aromatics, aldehydes and ketones produced by pyrolysis in the N$_2$ atmosphere were higher. When the heating rate was 20 K/min, the comprehensive combustion parameters were 56.442 and 6.871 × 10$^{-7}$%$^2$/(min$^2$• K$^3$) in the air and N$_2$ atmospheres, respectively, indicating that the peanut shells had great potential to become bioenergy.

**Keywords:** peanut shells; biomass; kinetic reaction model; kinetic transformation; thermodynamic transformation

## 1. Introduction

The intense change of global climate is associated with the increasing of the country's consumption of fossil fuels. The limited availability of fossil fuels and their rate of consumption have led to the widespread development of alternatives to renewable energy. As a renewable and carbon dioxide neutral energy source, biomass fuels have received increasing attention in recent years [1,2]. It is reported that biofuels are not only a renewable energy source, but also reduce carbon dioxide, nitrogen oxides, hydroxides and sulfur oxide emissions [3–6]. Biomass energy has become a replacement for 14% of global energy consumption. There have been various studies on biomass pyrolysis, focusing on waste mushroom matrix [7], straw [8], water hyacinth [9], tea residue [10], bagasse [3] and coffee grounds [11].

The annual production of the peanut shells (PSH) exceeds 5 million tons in China, the country with the largest peanut production in the world. Even if a certain amount of the PSH are consumed in the production of animal feed, most of the PSH are discarded, resulting in serious environmental

problems [12]. Therefore, there is an urgent need for a method to properly treat the PSH. Gasification, pyrolysis and combustion are the latest thermochemical processes [13], while coke formation, depolymerization, fracture and secondary reactions are the main reactions in the pyrolysis of biomass. At present, the main direction of solid waste treatment remains on pyrolysis and combustion.

The feasibility or applicability of any bioenergy generation depends on a range of parameters such as the physicochemical properties of the feedstock, heating rate, pyrolysis temperature, type of atmosphere and kinetics and thermodynamics. In order to quantify the above conditions, the pyrolysis and combustion process of the PSH were analyzed by TG combined with FTIR and the kinetic reaction models FWO and DEAM. Recently, by-products and energy produced by the (co)pyrolysis of biomass in an oxygen-rich and non-oxygen-rich environment have begun to receive much attention. Biomass rich in lignocellulose was considered to have great energy production prospects [14]. For example, Kayahan and Ozdogan revealed the state of $SO_2$ and NO emissions from the co-combustion of biomass and lignite in an oxygen-rich fluidized bed [15]. Meng et al. found that the influences of the air ($N_2/O_2$) and oxyfuel ($CO_2/O_2$) atmospheres on the properties and combustion characteristics of coal gangue were different [16]. Li et al. placed two typical biomasses in two coastal areas in a nitrogen atmosphere for pyrolysis experiments [17]. Bio-oil, which is considered a value-added by-product, is concentrated from devolatilized pyrolysis gas. [18]. For the purpose of heating it more quickly, Xu et al. jointly pyrolyzed seaweed and rice husk in $N_2$ atmosphere in the pyrolysis reactor to explore its mechanism [19]. Cheng et al. found that some negative effects on biomass combustion performance will occur if $CO_2$ is used instead of $N_2$, such as combustion stability, ignition, gas temperature, and heat transfer [20]. As the simplest way to convert biomass that is widely practiced commercially, combustion can provide heat and power, but its low conversion efficiency makes this method is not environmentally friendly. Because bio-oil can be highly oxidized and its chemical properties are unstable, reducing the oxygen content or removing waste is a matter that must be paid attention to in the process of obtaining bio-oil. Therefore, this study selected air and $N_2$ atmosphere to explore the pyrolysis and combustion characteristics of the PSH.

The objectives of this study were to (1) use the TG experiments to quantify the pyrolysis and combustion characteristics of the PSH in two different atmospheres; (2) estimate the activation energy (*Ea*) by utilizing FWO and DEAM model methods; (3) determine the performances and thermodynamic parameters; (4) determine the functional group of the gas by using TG in combination with FTIR.

## 2. Material and Methods

### 2.1. Sample Preparation

Peanuts were obtained from the village street Xinjie vegetable market in Guangzhou, China. In the pretreatment stage, the peanuts were dehulled after washing with deionized water, and the obtained PSH were washed three times with deionized water and then allowed to dry naturally. The purpose of holding in the oven at 105 °C for 24 h is to further reduce the moisture in the sample. After grinding below 200 mesh, the dryer was used to hold the sample.

The "China Solid Biofuels (GB/T28731-2012)" was used to analyze their ash (A), moisture (M) and volatile matter ($V_M$) contents. The final analysis was performed using an elemental analyzer (Vario EL cube by Elementar) and the approximate, final results and calorific value analysis for the PSH are listed in Table 1.

**Table 1.** Ultimate analyses, proximate analyses and high heating value.

| Biomass | Ultimate Analysis (wt.%) | | | | | Proximate Analysis (wt.%) | | | | HHV (MJ/kg) | H/C | O/C |
|---|---|---|---|---|---|---|---|---|---|---|---|---|
| | C | O | H | S | N | M | FC | A | $V_M$ | | | |
| Peanut | 43.55 | 34.54 | 5.4 | 0.06 | 1.52 | 7.19 | 17.80 | 7.83 | 67.18 | 19.17 | 0.12 | 0.79 |

## 2.2. Thermogravimetric Experiments

TG/DSC analysis was performed using a TG analyzer (NETZSCHSTA 409 PC). The samples weighing 5.0 ± 0.5 mg were placed in alumina crucibles and then warmed to 900 °C using the previously selected the heating rates of 5, 10, 20 and 40 K/min. Data on mass loss and heat flow were recorded by analyzer software. Changing the atmosphere or heating rate required re-establishing a blank experiment, and subtracting the systematic error caused by the blank when loading the sample experiment, and repeating the experiment three times to ensure that the repeatability of the experiment was maintained within the error range of 2%.

## 2.3. TG-FTIR Analyses

A TG analyzer (NETZSCHSTA 449 F3, Germany) and FTIR (NICOLET iS50 FTIR, USA) were applied to analyze the functional groups of gaseous products continuously produced by the PSH during pyrolysis and combustion. The heating rate of 20 K/min was selected and 5 ± 0.5 mg of the same weight peanut shell was heated in the air and nitrogen atmosphere. All samples started to warm up at the room temperature and the termination temperature was selected to be 900 °C. To prevent the gas from condensing during the process of entering the equipment, all samples were transported at 280 °C and then spectrally sampled at a resolution of 4 $cm^{-1}$. The resolution range is 400 $cm^{-1}$–4000 $cm^{-1}$ and each experiment requires a blank experiment in advance to eliminate the background signal.

## 2.4. Pyrolysis Performance Indices

The following four pyrolysis performance indicators were selected: ignition ($D_i$), combined combustion ($S$), flammability ($C$) and burnout ($D_b$). In addition, the study used the following five parameters required to estimate thermal behavior: maximum mass loss rate ($-R_p$), average mass loss rate ($-R_V$), $T_i$, $T_b$ and the time range of half values of $-R_p$ ($\Delta t_{1/2}$) [7,21]. The extent of overall combustion performance and combustion potential can be reflected by $S$ and $D_b$, respectively. The four pyrolysis performance indicators are as follows [22,23]:

$$S = \frac{(-R_p) \times (-R_v)}{T_i^2 \times T_b} \tag{1}$$

$$D_i = \frac{(-R_p)}{t_i \times t_p} \tag{2}$$

$$D_b = \frac{(-R_p)}{\Delta t_{1/2} \times t_p \times t_b} \tag{3}$$

$$C = \frac{(-R_p)}{T_i^2} \tag{4}$$

where $t_i$, $t_b$ and $t_p$ represent the time of ignition, burnout and peak, respectively. Better pyrolysis efficiency is generally indicated by lower ignition and higher burnout index.

## 2.5. Kinetic Analyses

The kinetic parameters of oxidative and thermal degradation processes were used to analyze the experimental results of TG [7].

$$\frac{d\alpha}{dt} = k(T) \times f(\alpha) \tag{5}$$

The conversion degree ($\alpha$) and the reaction rate constant $k$ $(T)$ can be calculated by the following formula:

$$k(T) = A \exp(-\frac{E_\alpha}{RT}) \tag{6}$$

$$\alpha = \frac{m_0 - m_t}{m_0 - m_\infty} \tag{7}$$

The final residual mass, the initial mass and the instant mass of the sample are represented by $m_\infty$, $m_0$ and $m_t$, respectively. The value of the universal gas constant is 8.314 J/mol/K, which is represented by $R$. $\beta = dT/dt$, represents a constant heating rate and Equations (5) and (8) can be converted to each other [11].

$$\frac{d\alpha}{dT} = \frac{A}{\beta} e^{-(E\alpha/RT)} \times f(\alpha) \tag{8}$$

Two non-isothermal methods, DAEM and FWO were selected for comparative kinetic analysis. As a model for simulating complex pyrolysis processes, DAEM is not only effective and accurate, but also has many first-order parallel irreversible reactions [24]. It is assumed that errors caused by function mechanism functions can be effectively avoided by using FWO, thus, it is often used to directly find the activation energy [25–27].

$$FWO : \lg\beta = \lg(\frac{E_\alpha}{RG(\alpha)}) - 2.315 - 0.4567\frac{E_\alpha}{RT} \tag{9}$$

$$DEAM : \ln(\frac{\beta}{T^2}) = \ln(\frac{R}{E_\alpha}) + 0.6575 - \frac{E_\alpha}{RT} \tag{10}$$

## 3. Results and Discussion

### 3.1. Chemical Properties and Basic Physical of the PSH

The ultimate and proximate analysis results were listed in Table 1. Based on its relatively low sulfur content (Table 1), the emissions of sulfur oxides produced by the PSH during pyrolysis will be low, which reflected that the burning of the PSH seemed to be compatible with the environment. Higher thermal efficiency can be reflected by lower oxygen to carbon ratio and hydrogen to carbon ratio [28]. Compared to rice husks (0.10 and 0.84, respectively) [29], a relatively lower O/C ratio and the higher H/C ratio can be derived from the physicochemical properties of the peanut shell, so that the PSH were more suitable for burning than rice husks.

The ignition performance of the fuel was related to the sum of fixed carbon and volatiles. The higher sum of fixed carbon and volatiles indicated the fuel had a better ignition performance [7]. The sum of volatile matter and fixed carbon of the PSH (84.98%) was higher than water hyacinth (72.65%) [9], indicating a better ignition performance of the PSH. Determining whether more oxygen-containing functional groups are included in volatile organic compounds can be derived from high O/C and low H/O [30]. The calorific value (HHV) of PSH (19.17 MJ/kg$^{-1}$) was about twice that of raw coal (10.56 MJ/kg$^{-1}$) [31], making the PSH a promising candidate for bioenergy production.

### 3.2. The Analysis of Thermogravimetric

#### 3.2.1. Decomposition Behavior at 20 K/min

Figure 1 was the DTG curve when the heating rate was 20 K/min. The evaporation of water, the pyrolysis of volatiles, the volatiles of the residues and the pyrolysis of the fixed carbon correspond to the three peaks of DTG in $N_2$ atmosphere. Unlike in the $N_2$ atmosphere, the PSH were burned in the air atmosphere in four stages, and the volatiles of the residue and the pyrolysis of the fixed carbon were separated into two stages.

The TG curve decreased slightly before 200 °C and dropped sharply until steady state was reached (Figure 1). In the two atmospheres, the DTG curve showed that the PSH burned with three similar peaks, two of which were slight peaks and the other was a slight peak in the air atmosphere (Figure 1). The weight loss of the first stage in the air and $N_2$ atmosphere was 5.27% and 7.24%, respectively, which referred to the evaporation of moisture from the surface of the PSH during

pyrolysis from the room temperature to 150 °C. The PSH contain a large amount of lignocellulose. When the sample water content is less than 10%, they are generally suitable for combustion [32]. The initial start of depolymerization and volatile pyrolysis mainly appeared in the second stage with the cracking of small amount of low-polymerization hemicellulose and the polymerization reaction of some cellulose [33]. The maximum mass loss of the PSH in the two atmospheres occurred in the third stage, the depolymerization of the PSH and the pyrolysis of volatiles were considered to continue with the decomposing of lignin at the same stage. Maximum mass loss rate of the PSH in the air and $N_2$ atmospheres were about 45.89 and 11.64%/min, respectively. The temperature at the peaking of the PSH in the air atmosphere was 289 °C, which was lower than the peak temperature in the $N_2$ atmosphere. The pyrolysis of fixed carbon and residual lignin was the last phase before the end of the pyrolysis of the PSH in the air atmosphere.

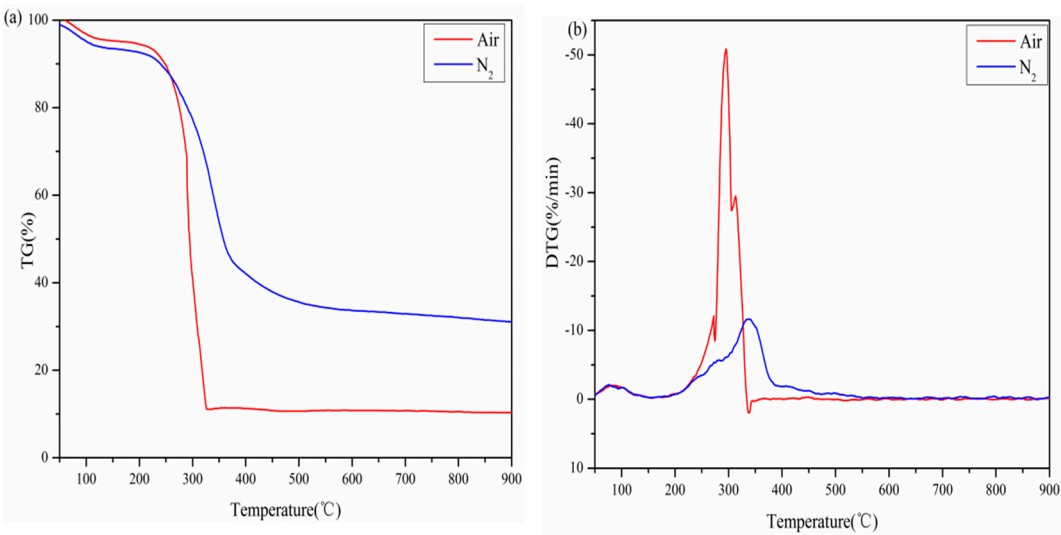

**Figure 1.** (**a**) TG and (**b**) DTG curves of the PSH at the heating of 20 K/min in two atmospheres.

### 3.2.2. Influence of Different Heating Rate

A similar decomposition pattern in both atmospheres can be seen from the (D)TG curve in Figure 2, but the apparent differences in peak height indicated that the burning of the PSH were affected by the heating rate. The combustible reaction was substantially completed when the temperature reached 500 °C in the air atmosphere, while there was still a small amount of decomposition of combustibles in the final stage of the PSH in the $N_2$ atmosphere. DTG curves tended to move back in the $N_2$ atmosphere when the heating rate increasing, but the residual amount of the PSH in the two atmospheres showed no change substantially with the increase of the temperature.

For the same temperature and sample, the degree of sample reaction and decomposition increased with decreasing heating rate [34]. As can be seen from Figure 2, the PSH decomposed better during the low heating rates. During each heating rate, maximum mass loss (4.21%/min–74.73%/min) and average mass loss rate (0.55%/min–5.55%/min) of the PSH in the air atmosphere were higher than mass loss rate (2.32%/min–28.15%/min) and average mass loss rate (0.32%/min–4.06%/min) in the $N_2$ atmosphere, so that the pyrolysis in air was more intense than in the $N_2$ atmosphere. At all heating rates, the burning time of the PSH in the air was lower than the burning time in $N_2$ for the release of volatiles may be accelerated and the possibility of coking of intermediates may be reduced [35]. Part of the time were used for the pyrolysis of volatiles, and a certain temperature difference inside and outside the sample can also have a negative impact on thermal decomposition, while heat transfer can be hindered as the heating rate increases [36]. Bio-oil is the main product of pyrolysis, and the short residence time of the vapor, the moderate temperature and the higher heating rate are more favorable for bio-oil formation [37]. Considering that the heating rate and temperature have an influence on the pyrolysis

process, choosing a suitable temperature and a higher heating rate under a nitrogen atmosphere may be beneficial to the production of bio-oil.

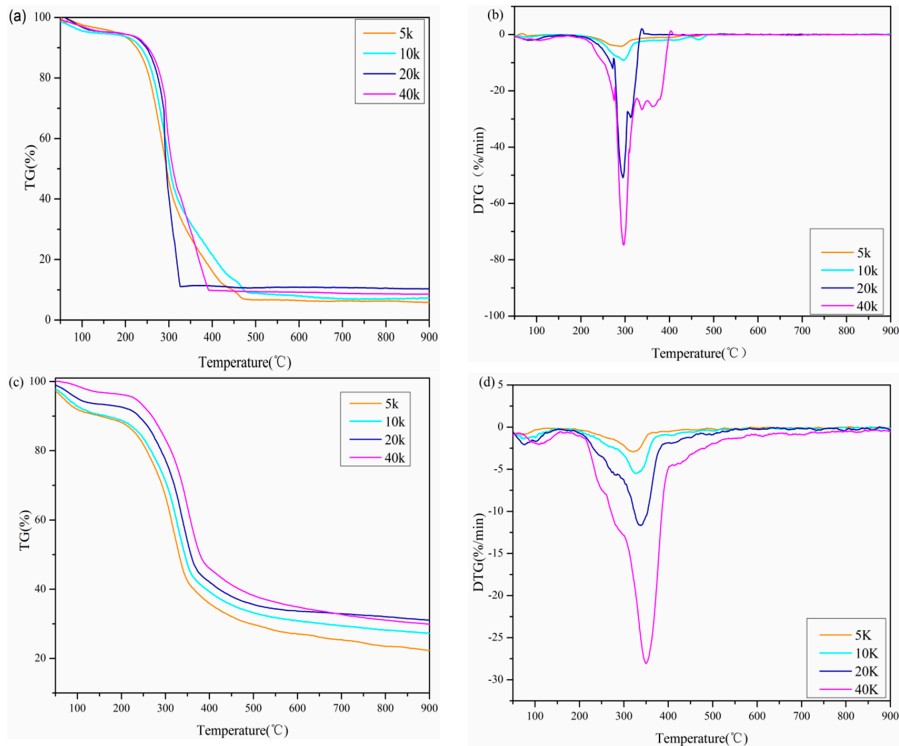

**Figure 2.** (D)TG curves of the PSH pyrolysis in the (**a**) and (**b**) air and (**c**) and (**d**) $N_2$ atmosphere.

In the air and $N_2$ atmosphere, $T_i$ and $T_p$ gradually moved to a higher temperature region when the heating rate increased. A lower value of $D_i$ and higher values of $S$, $C$ and $D_b$ indicated a better combustion efficiency. After calculating $S$, $C$, $D_i$ and $D_b$ of the PSH, it was found that the four values showed similar trends in the air and $N_2$ atmospheres. The values of $S$, $C$, $D_i$ and $D_b$ both increased sharply when the heating rate increased. The difference is that the change in heating rate does not cause the values of $C$, $S$ and $D_b$ in air to be higher than in nitrogen, which indicated that the PSH were more efficient in the air atmosphere. The opposite phenomenon occurred in the $N_2$ atmosphere with a lower $D_i$ value (Table 2). Therefore, a comprehensive comparison showed that a high heating rate in air atmosphere may be more favorable for pyrolysis decomposition of the PSH. Compared with the $S$ value of coal ($8.39 \times 10^{-9}\%^2/(\text{min}^2 \cdot \text{K}^3)$) [31], the $S$ value of the PSH was higher than the $S$ value of coal, which provided a basis for the PSH as a better biofuel.

**Table 2.** Pyrolysis characteristic parameters of the PSH based on TG experiments.

| Biomass | $\beta$ | Temperature (°C) | | | $-R_V$ | $-R_P$ | Time (Min) | | Pyrolysis Parameters | | | |
|---------|---------|-------|-------|-------|--------|--------|--------|--------|---------|---------|---------|---------|
| | | $T_i$ | $T_p$ | $T_b$ | | | $t_p$ | $t_b$ | $S$ | $C$ | $D_i$ | $D_b$ |
| Air | 5 | 236 | 289 | 346 | 4.21 | 0.55 | 46.70 | 59.43 | 1.201 | 7.559 | 0.242 | 0.011 |
| | 10 | 252 | 293 | 348 | 9.09 | 1.07 | 25.34 | 31.64 | 4.401 | 14.314 | 1.644 | 0.204 |
| | 20 | 269 | 295 | 314 | 50.89 | 2.52 | 13.12 | 14.01 | 56.442 | 70.328 | 31.767 | 18.335 |
| | 40 | 270 | 297 | 338 | 74.73 | 5.55 | 7.11 | 7.63 | 168.323 | 102.51 | 164.742 | 270.104 |
| $N_2$ | 5 | 242 | 316 | 384 | 2.93 | 0.46 | 55.38 | 68.99 | 0.594 | 5.003 | 0.130 | 0.005 |
| | 10 | 256 | 331 | 390 | 5.49 | 0.84 | 29.18 | 35.12 | 1.804 | 8.377 | 0.872 | 0.078 |
| | 20 | 268 | 336 | 400 | 11.68 | 1.69 | 14.71 | 17.94 | 6.871 | 16.262 | 6.881 | 0.790 |
| | 40 | 281 | 350 | 406 | 28.05 | 3.91 | 8.39 | 9.58 | 34.211 | 35.524 | 47.355 | 24.235 |

$t_p$ and $t_b$: peak and burnout time, min; $\beta$: heating rate, K/min; $T_i$, $T_b$ and $T_p$: ignition, burnout and peak temperature, °C; $-R_V$ and $-R_P$: average and maximum mass loss rate, %/min; $C$: flammability index, $10^{-5}\%/(\text{min} \cdot \text{K}^2)$; $S$: comprehensive combustion index, $10^{-7}\%^2/(\text{min}^2 \cdot \text{K}^3)$; $D_b$: degree of possibility of combustion, $10^{-2}\%/\text{min}^4$. $D_i$: Ignition index, $10^{-2}\%/\text{min}^3$.

### 3.3. DSC Analyses

Analysis of the exothermic and endothermic phases during pyrolysis of the PSH required the aid of DSC curves, and the entire pyrolysis stages of the DSC curve with four heating rates were presented in the Figure 3. It can be seen that the PSH were basically burned in the air and $N_2$ atmospheres. This was an exothermic process and the exothermic effect gradually increased when higher heating rates occurred. The DSC curve at 20 K/min in the air atmosphere was stronger than the exothermic effect at 40 K/min, which may be due to experimental errors due to high heating rates [38].

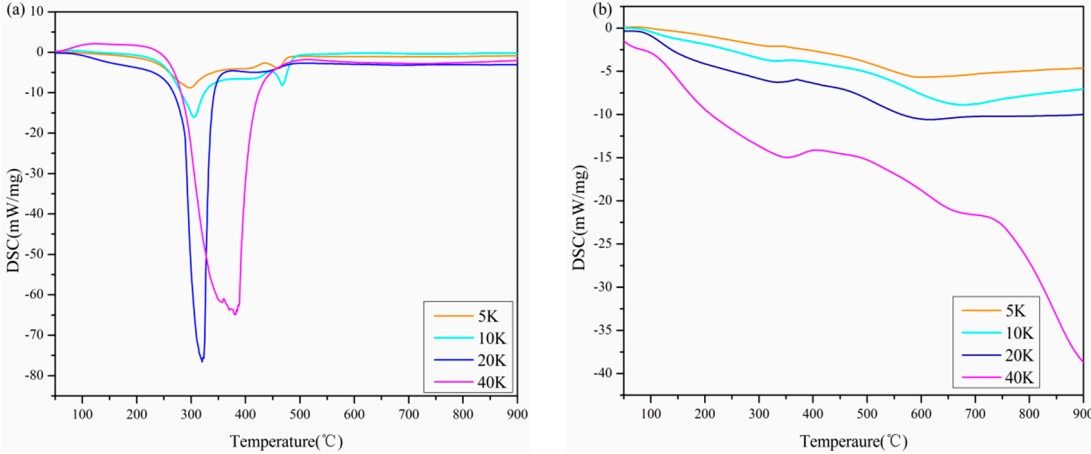

**Figure 3.** DSC curves of the PSH pyrolysis in the (**a**) air and (**b**) $N_2$ atmosphere at four heating rates.

In the air atmosphere, two peaks can be observed in the DSC curve and the more severe peak appeared above 300 °C corresponding to the process of volatile pyrolysis in the DTG curve. As the heating rate increased, the peak of the DSC curve appeared for a longer time than the peak of the DTG. This may be due to the presence of a certain degree of hysteresis between the outer and inner particles that hindered the release of heat. After 500 °C, the heat flow on the DTG curve began to approach zero; then, the DTG curve maintained a substantially constant state, indicating that the sample weight was essentially unchanged which was also substantially consistent with the TG curve. In the $N_2$ atmosphere, the curve of the DSC was significantly different from that in the air atmosphere for the peaks were not intense. The DSC curve in the $N_2$ atmosphere contained three peaks, which was consistent with the DTG curve and the related mass loss that can be obtained from the curve of TG. The heat flux in the second and third phases changed significantly and it can be clearly seen on the DTG curve that the two peaks differed a lot. That may be due to the decomposition of fixed carbon and volatiles in the two phases, and volatile matter involved more carbon-oxygen bonds than FC, while higher energy exists in carbon-carbon bonds instead of carbon-oxygen bonds [36].

### 3.4. Kinetic Analyses

The energy that a molecule acquires from a normal state to a sensitive state is called activation energy, which is the most basic indicator of ease of reaction. The lower the Ea value, the easier the reaction will occur. The peanut shell samples are rich in (semi)cellulose and lignin, and the Ea values of pyrolysis of amorphous lignin and hemicellulose are lower than the activation energy of cellulose pyrolysis [39]. FWO and DEAM models were used in this study to estimate the *Ea* value of the PSH burning in both atmospheres. Figure 4 showed that the activation energy in the air atmosphere obtained by the FWO or DEAM models were higher than the activation energy in the $N_2$ atmosphere, this is because the substance had a lower porosity for $N_2$ molecules than $CO_2$ molecules [40,41]. The higher value of $R^2$ indicated that both models can accurately calculate the Ea value, but their accuracy and applicability are different. FWO achieved a higher $R^2$ value (0.9639) in air, while DEAM achieved a higher $R^2$ value (0.9330) in $N_2$. The average values of $R^2$ obtained by estimating the *Ea* value by FWO

and DEAM in the air atmosphere were 0.9639 and 0.9215, respectively, and the average values in the $N_2$ atmosphere were 0.9531 and 0.9330, respectively (Table 3).

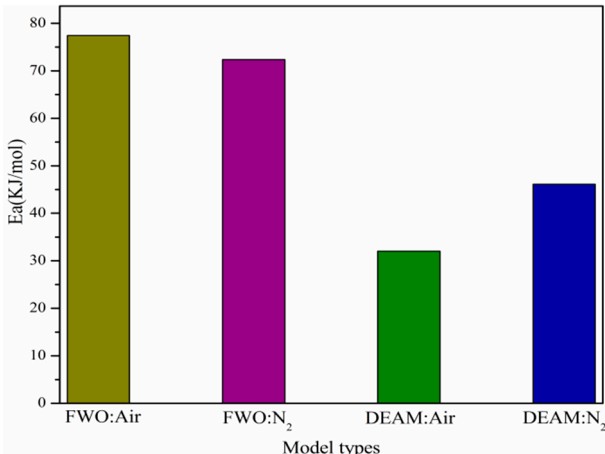

**Figure 4.** The average value of *Ea* calculated using FWO and DEAM in two atmospheres.

**Table 3.** The values of $R^2$ and Ea according to FWO and DAEM at 20K•min$^{-1}$.

| | FWO | | | | DAEM | | | |
|---|---|---|---|---|---|---|---|---|
| | **Air** | | **N$_2$** | | **Air** | | **N$_2$** | |
| *a* | *Ea* | *R$^2$* | *Ea* | *R$^2$* | *Ea* | *R$^2$* | *Ea* | *R$^2$* |
| 0.10 | 36.34 | 0.9966 | 2.54 | 0.9911 | 32.06 | 0.9703 | 0.24 | 0.9970 |
| 0.15 | 42.81 | 0.9966 | 11.89 | 0.9911 | 38.61 | 0.9818 | 2.70 | 0.9816 |
| 0.20 | 44.35 | 0.997 | 20.72 | 0.9902 | 41.67 | 0.9949 | 9.10 | 0.9978 |
| 0.25 | 48.42 | 0.9933 | 25.88 | 0.9741 | 45.47 | 0.9903 | 20.71 | 0.8392 |
| 0.30 | 53.58 | 0.9993 | 27.00 | 0.9308 | 50.37 | 0.9983 | 23.40 | 0.9996 |
| 0.35 | 70.07 | 0.9861 | 32.63 | 0.9392 | 62.97 | 0.9698 | 32.77 | 0.9995 |
| 0.40 | 85.6 | 0.9990 | 38.02 | 0.9503 | 81.26 | 0.9513 | 42.67 | 0.9998 |
| 0.45 | 97.99 | 0.9205 | 41.53 | 0.9601 | 92.43 | 0.9178 | 51.27 | 1.0000 |
| 0.50 | 117.74 | 0.9142 | 44.67 | 0.9556 | 111.04 | 0.9114 | 60.10 | 0.9998 |
| 0.55 | 117.03 | 0.9752 | 46.88 | 0.9568 | 110.46 | 0.9737 | 68.33 | 0.9999 |
| 0.60 | 131.31 | 0.9600 | 48.11 | 0.9455 | 123.90 | 0.9581 | 76.81 | 0.9986 |
| 0.65 | 116.95 | 0.9211 | 48.91 | 0.9696 | 110.30 | 0.9184 | 85.61 | 0.9919 |
| 0.70 | 151.72 | 0.8566 | 36.51 | 0.93 | 143.02 | 0.8531 | 19.87 | 0.8988 |
| 0.75 | 35.98 | 0.9824 | 25.86 | 0.8588 | 33.09 | 0.9807 | 90.07 | 0.9668 |
| 0.80 | 43.37 | 0.9992 | 34.08 | 0.8296 | 39.94 | 0.9996 | 68.06 | 0.7864 |
| 0.85 | 54.25 | 0.9775 | 32.09 | 0.777 | 50.00 | 0.9194 | 72.67 | 0.6422 |
| 0.90 | 69.22 | 0.9109 | 26.95 | 0.7158 | 63.86 | 0.9146 | 60.23 | 0.7626 |
| Average | 77.45 | 0.9639 | 32.01 | 0.9215 | 72.38 | 0.9531 | 46.15 | 0.9330 |

The change in the value of Ea calculated by FWO and DEAM in both atmospheres can be divided into two phases. In the different atmospheres, *Ea* calculated by the two models were basically in a gradually increasing trend when $\alpha$ was between 0.1 and 0.65. In the air atmosphere, when $\alpha$ was between 0.4 and 0.65, the stable values of *Ea* were displayed (100.0 ± 20 kJ/mol), showing a relatively slow rising trend, and the complexity of the middle reaction mechanism was largely due to the decomposition of (semi)cellulose [30]. In the $N_2$ atmosphere, when $\alpha$ was between 0.35 and 0.55, the *Ea* values (50.0 ± 20 kJ/mol) were the same as the steady trend in the air. The main decomposition phase can correspond to a relatively stable *Ea* value. In the second stage, the *Ea* values in the two atmospheres were not stable, the release of volatiles during pyrolysis led to the kinetic complexity of the different reactions, which can be seen by the broader *Ea* values. There was a significant downward trend in the second phase of the two different atmospheres; this may be due to the large correlation between the decomposition of fixed carbon during pyrolysis of the PSH. Many of the weak bonds

present in the sample molecule will produce less energy when broken, and these can be obtained from lower *Ea* values [42]. Therefore, the PSH exhibited great co-combustion feasibility.

### 3.5. Thermodynamic Analyses

For the purpose of understanding the burning of the PSH in two atmospheres, the thermodynamic parameters $\Delta S$, $\Delta G$, $\Delta H$ and $A$ were quoted, which can be calculated from the *Ea* values derived from the FWO and DEAM models (Table 4). Optimizing biomass combustion usually required changing the frequency of collisions between reactants, which can be explained by pre-exponential factors [43]. The exchanging of heat between the reagent and the activation complex can be reflected by the values of $\Delta H$ and $\Delta G$, respectively [9]. $\Delta S$ is an index used to measure the diseases that were associated with the formation of complex activated species [9].

**Table 4.** Thermodynamic parameters for the PSH pyrolysis using FWO at 20 K/min.

| *a* | $A$ (s$^{-1}$) | $\Delta H$ (kJ/mol) | $\Delta G$ (kJ/mol) | $\Delta S$ (J/mol) | $A$ (s$^{-1}$) | $\Delta H$ (kJ/mol) | $\Delta G$ (kJ/mol) | $\Delta S$ (J/mol) |
|---|---|---|---|---|---|---|---|---|
| 0.10 | $2.73 \times 10^6$ | 34.33 | 72.21 | −128.42 | $1.34 \times 10^{-1}$ | 1.04 | 90.78 | −226.39 |
| 0.15 | $4.51 \times 10^7$ | 40.64 | 71.81 | −105.65 | $1.79 \times 10^1$ | 9.88 | 86.46 | −197.04 |
| 0.20 | $8.76 \times 10^7$ | 42.10 | 71.72 | −100.42 | $7.34 \times 10^2$ | 18.51 | 84.91 | −180.40 |
| 0.25 | $5.01 \times 10^8$ | 46.10 | 71.51 | −86.13 | $5.82 \times 10^3$ | 23.55 | 84.29 | −177.06 |
| 0.30 | $4.55 \times 10^9$ | 51.22 | 71.26 | −67.94 | $9.08 \times 10^3$ | 24.57 | 84.17 | −159.07 |
| 0.35 | $4.94 \times 10^{12}$ | 67.66 | 70.60 | −9.96 | $8.22 \times 10^4$ | 30.09 | 83.64 | −142.05 |
| 0.40 | $3.40 \times 10^{15}$ | 83.19 | 70.11 | 44.33 | $6.60 \times 10^5$ | 35.40 | 83.21 | −131.12 |
| 0.45 | $6.09 \times 10^{17}$ | 95.57 | 69.78 | 87.44 | $2.53 \times 10^6$ | 38.84 | 82.97 | −121.37 |
| 0.50 | $2.29 \times 10^{21}$ | 115.31 | 69.33 | 155.87 | $8.37 \times 10^6$ | 41.92 | 82.76 | −114.56 |
| 0.55 | $1.71 \times 10^{21}$ | 114.59 | 69.34 | 153.37 | $1.94 \times 10^7$ | 44.09 | 82.63 | −110.84 |
| 0.60 | $6.48 \times 10^{23}$ | 128.85 | 69.06 | 202.67 | $3.09 \times 10^7$ | 45.26 | 82.56 | −108.46 |
| 0.65 | $1.65 \times 10^{21}$ | 114.46 | 69.35 | 152.94 | $4.19 \times 10^7$ | 46.02 | 82.51 | −147.95 |
| 0.70 | $3.07 \times 10^{27}$ | 148.70 | 68.71 | 271.16 | $3.69 \times 10^5$ | 33.56 | 83.33 | −182.69 |
| 0.75 | $2.33 \times 10^6$ | 32.81 | 72.24 | −133.66 | $5.77 \times 10^3$ | 22.84 | 84.29 | −156.11 |
| 0.80 | $5.73 \times 10^7$ | 40.08 | 71.78 | −107.46 | $1.45 \times 10^5$ | 30.94 | 83.52 | −162.92 |
| 0.85 | $6.06 \times 10^9$ | 50.83 | 71.23 | −69.16 | $6.66 \times 10^4$ | 28.68 | 83.69 | −180.45 |
| 0.90 | $3.45 \times 10^{12}$ | 65.65 | 70.63 | −16.89 | $8.90 \times 10^3$ | 23.16 | 84.17 | −181.59 |
| Average | $1.81 \times 10^{26}$ | 74.83 | 70.63 | 14.24 | $6.14 \times 10^6$ | 29.31 | 84.11 | −157.65 |

The $A$ value in two different atmospheres varied with $\alpha$. The $A$ values in the N$_2$ atmosphere ($1.34 \times 10^{-1}$–$6.14 \times 10^6$) were less than $10^9$ s$^{-1}$ and fluctuated between $2.33 \times 10^6$–$3.07 \times 10^{27}$ in the air atmosphere. $A$ simple complex reaction or the end of complex reaction (or a surface reaction) can be displayed by the $A$ values ($10^9$ s$^{-1}$) [44]. A lot of complex chemical reactions may occur during the pyrolysis of the PSH at two atmospheres and cause strong chemical bonds in the intermediate stages of the decomposition process to be destroyed. The mean values of $\Delta H$ for the air and N$_2$ atmosphere were 74.83 and 29.31, respectively. 3 kJ /mol was the only small difference between the values of $\Delta H$ and *Ea* in two atmospheres, which represented the ease of product formation [45]. The $\Delta G$ value under different atmospheres fluctuated with the change of $\alpha$, and the mean value of $\Delta G$ was less than 15 kJ / mol in two atmospheres. It was noteworthy that large $\Delta G$ values occurred at the beginning and end of the reaction, indicating that too much thermal energy was supplied to the system at the beginning and end of the pyrolysis [45]. Low entropy values indicate that the thermodynamic equilibrium is closed to a low activity response [44]. The average value of $\Delta S$ in N$_2$ was lower than that of air, which indicated that the reactivity of the higher reactants can be obtained in air.

### 3.6. The Analyses of TG-FTIR

The three-dimensional (3D) TG-FTIR spectrum of the gas released by the PSH after burning in the air and N$_2$ atmosphere at 20 K/min were shown in Figure 5. Several absorption peaks were

apparently located near 700, 1000, 1500, 1700 and 2500 cm$^{-1}$. In the air atmosphere, the absorption peak at 2400 cm$^{-1}$ was most obvious, and all the peak shapes tended to disappear after 25 min, which corresponded to the decomposition of the PSH in the TG curve after the decomposition of 500 °C. In nitrogen, there were obvious absorption peaks at 1700 and 2400 cm$^{-1}$, and some irregular peak shapes existed in the late pyrolysis stage, which were consistent with the weak downward trend of the TG curve. The volatiles were released by using the analysis of TG-FTIR, with the chosen time corresponding to the peak of the ignition phase, the maximum and final burnout phases (Figure 6).

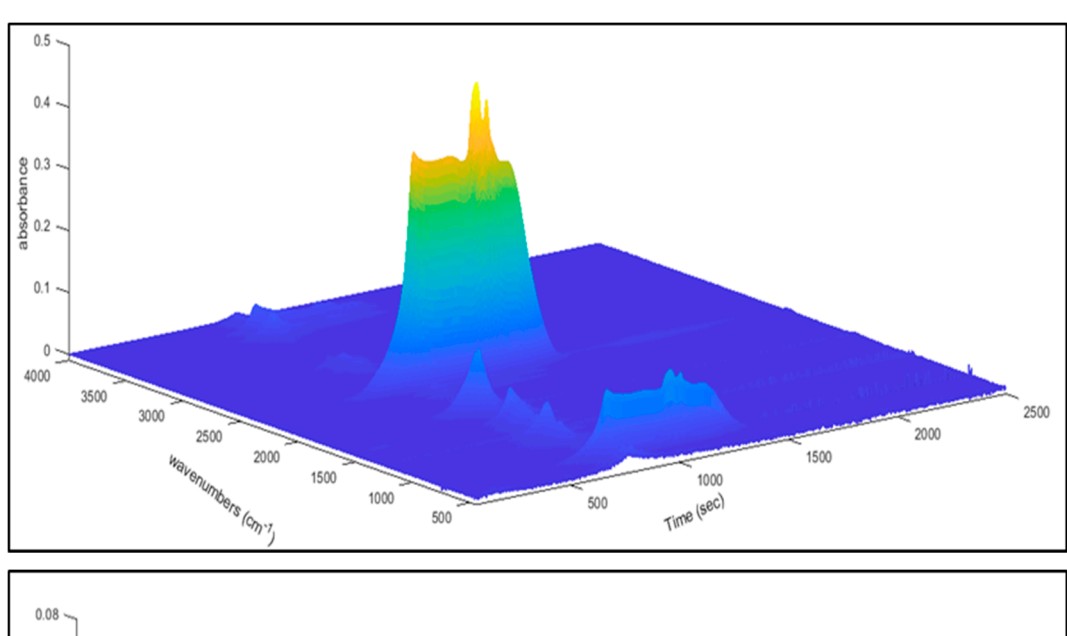

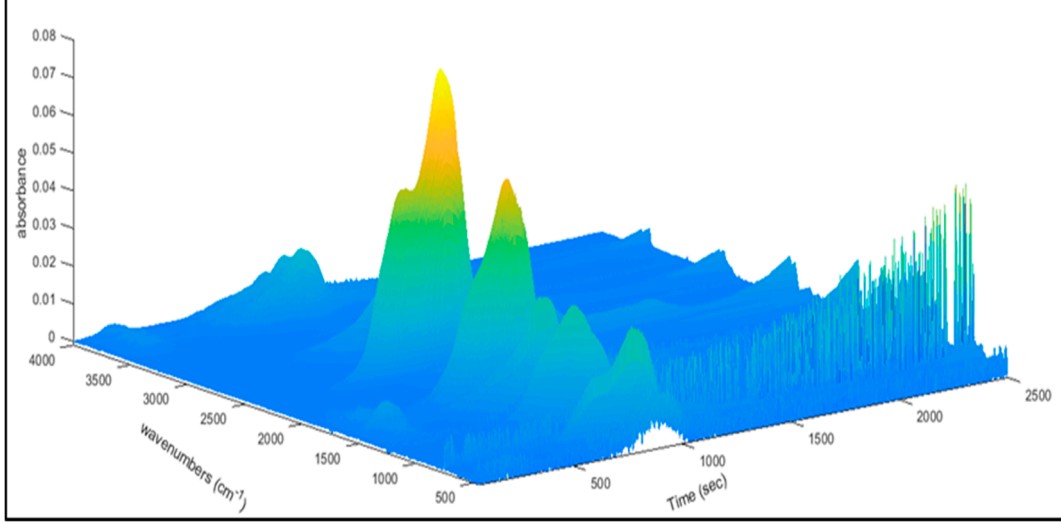

**Figure 5.** Three-dimensional (3D) infrared spectrum of gaseous products from the PSH pyrolysis. In the (top) air and (bottom) N$_2$ atmosphere.

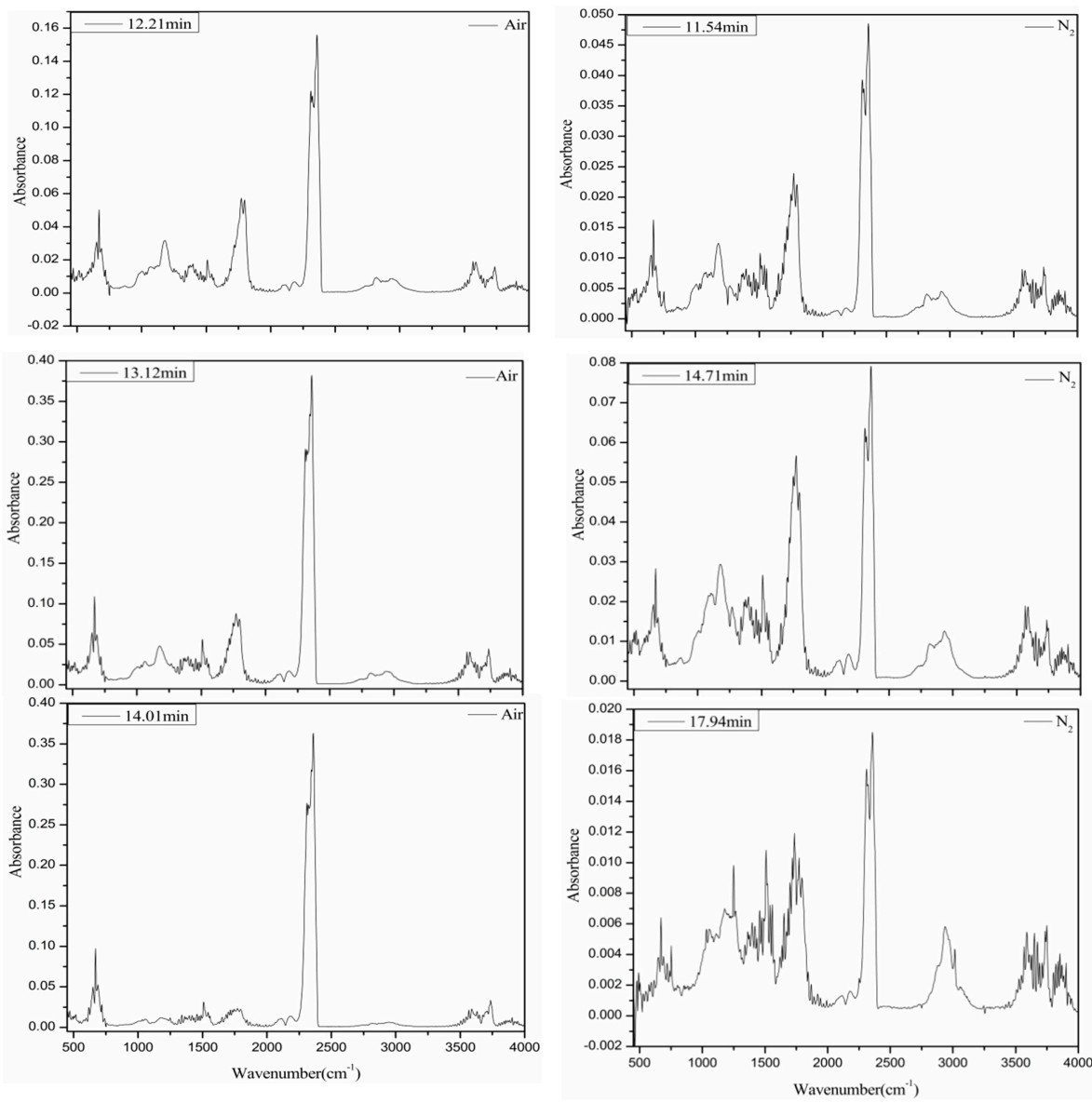

**Figure 6.** At three selected times in the air and $N_2$ atmosphere, respectively.

At 12.21 min, the O-H stretching vibration of 3500–3750 cm$^{-1}$ corresponded to the onset of decomposition of hemicellulose in the air atmosphere. Since O-H was mainly decomposed by water, this stage mainly corresponded to the evaporation of water. At 13.12 min, the gaseous product $CO_2$ was caused by the asymmetric stretching of C=O, thus producing absorbance at 2250–2400 cm$^{-1}$, at which time the absorption peak of $CO_2$ reached the maximum peak. At 1600–1800 cm$^{-1}$, the absorbance was due to the stretching of C=O, and the gaseous products obtained were mainly aldehydes and ketones. The absorbance at 600–700 cm$^{-1}$ was due to the release of aromatic gaseous products after C-H bending during the pyrolysis of the PSH. At 14.91 min, except for the increase in absorbance of aromatic substances at 600–700 cm$^{-1}$, the other absorbance showed a tendency to decrease. Asymmetric stretching of C=O and O-H stretching vibration occurred in 11.54 min, 14.71 min and 17.94 min in a nitrogen atmosphere, and C-O vibration and C-H stretching vibration appeared at 1000–1300 cm$^{-1}$ and 2750–3000 cm$^{-1}$, respectively. The linear relationship between the intensity of the absorption peak of the given gas and the concentration of the gas can be derived from Lambert-Beer's law [44]. Therefore, it can be seen from Figure 6 that the concentration of aromatic, aldehyde and ketone substances produced by the PSH in the $N_2$ atmosphere were higher.

## 4. Conclusions

A lagged peanut shells pyrolysis was caused by the increased heating rate, increasing its $T_i$ and $T_p$ to a higher temperature. However, the values of $C$, $S$ and $D_b$ in air atmosphere were higher than those in the $N_2$ atmosphere, indicating that the biomass were more efficient in pyrolysis in the air atmosphere. Both models obtained higher $R^2$ values when calculating the $Ea$ values, but their accuracy and applicability were different. Pyrolysis of peanut shells under the $N_2$ atmosphere at a suitable temperature and high heating rate may be beneficial to the production of bio-oil. The TG-FTIR results show that the absorption bands at the wave numbers were corresponded to C=O, C-O, C-H and O-H, and the gas evolution includes aldehydes, ketones and aromatic groups.

**Author Contributions:** Date curation: Z.X. and X.X.; formal analysis: L.Y. and J.H.; methodology: H.W.; project administration: Z.X. and P.F.; supervision: P.F. and Z.T.; validation: D.C.; writing—original draft: Z.X. All authors have read and agreed to the published version of the manuscript.

**Funding:** This work was supported by the Project of Science and Technology Program of Guangdong Province (2018B020208002), the National Natural Science Foundation of China (NSFC-51778264), the Central-Level Nonprofit Scientific Institutes for Basic R&D Operations (PM-zx703-201904-080), the youth Top-notch Talent Special Support Program of Guangdong Province (2016TQ03Z576), the Science and Technology Key Projects of Guangdong Province (2017A030223005), the Science and Technology Program of Guangzhou City (201804010147).

**Conflicts of Interest:** The authors declare no conflict of interest.

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
