# Peer review of "Comparison of Combustion and Pyrolysis Behavior of the Peanut Shells in Air and N2: Kinetics, Thermodynamics and Gas Emissions"

_sustainability, doi:10.3390/su12020464_

Round 1
Reviewer 1 Report
The considered manuscript is devoted to the study of one of the important biofuels production urgent problems - the conditions for pyrolysis optimization in the processing of plant waste using optimization on the example of peanut shells.
The authors used quite adequate methods (Differential Scanning Calorimetry, DSC; Thermogravimetric experiments) and models (FWO and DEAM).
The characteristics of the exothermic and endothermic pyrolysis phases of peanut shells were evaluated using DSC curves and other methods.
It was shown that peanut shells were effectively destroyed both in air and in the N2 atmosphere, but the kinetic pyrolysis curves were significantly different.
Numerous experimental data are presented in the manuscript and the discussion was carried out using modern literature - about half of all cited sources are publications of the last three years.
Thus, in my opinion, the submitted manuscript can be recommended for publication. However, there are also some comments to improve the manuscript.
1) In the «Introduction» section, the authors note that, according to available data, the use of pyrolysis in a nitrogen atmosphere can provide some advantages (in particular, «Bio-oil was concentrated from a devolatilized pyrolysis gas to be used as a value-added by-product [18]»). However, it seems important to somewhat detail this provision: to clarify the possible composition of the resulting «by-product», its output and prospects for use.
2) In section 3.2.2. and other sections of chapter 3, the authors provide convincing data on the differences in the pyrolysis curves carried out in atmospheric air and N2 atmosphere, however, the answer to the question posed in the introduction remains unclear: does replacing atmosphere air with N2 atmosphere ensure efficient «by-product» production.
3) It seems advisable to include in the «Conclusions», even in the most general form, considerations on the possible economic profitability of replacing atmosphere air with an N2 atmosphere using pyrolysis to produce biofuels and by-products.
4) In order to improve the article, it would be better to include in the manuscript decoding of abbreviations.
Author Response
Please see the attachment.
Special thanks to you for your good comments.

Reviewer 2 Report
In this paper, authors investigated both combustion and pyrolysis of peanut shells in air and N2 and compared the phenomena. The paper tackles an interesting topic in the relevant field. I recommend the article for acceptance; however, this article needs revision with below questions answering:
The abstract section needs more clarification; authors can give more emphasis on findings. Introduction section failed to provide a clear statement of the problem, the relevant literature on the subject, and the proposed approach or solution. This section needs major improvement with more discussion on the problem identification and knowledge gaps. More details are needed in experimental designing. Currently, this section is incomplete and doubtful for readers. The author discussed the outcomes of their study in a very well manner however all the figures have very limited readability. Should be highly improved. Furthermore, the author should recheck “Results and Discussion” very carefully with all corresponding figures. A complete Nomenclature section should include to helps readers. There are many typo errors by using math type. Authors need to fix the accurate font size.
Author Response
Please see the attachment
Due to some copy problems, all responses were put in the attachment.

Round 2
Reviewer 2 Report
It can be accepted in the present form.